# Autosomal Dominant Polycystic Kidney Disease: Extrarenal Involvement

**DOI:** 10.3390/ijms25052554

**Published:** 2024-02-22

**Authors:** Matteo Righini, Raul Mancini, Marco Busutti, Andrea Buscaroli

**Affiliations:** 1Nephrology and Dialysis Unit, Santa Maria delle Croci Hospital, AUSL Romagna, 48121 Ravenna, Italy; andrea.buscaroli@auslromagna.it; 2Nephrology, Dialysis and Transplantation Unit, IRCCS Azienda Ospedaliero Universitaria di Bologna, 40138 Bologna, Italy; raul.mancini@studio.unibo.it (R.M.); marco.busutti@aosp.bo.it (M.B.)

**Keywords:** ADPKD, ciliopathies, cystic kidney disease, genetic, extrarenal cystic involvement, PLD, intracranial aneurysms, bone disorders

## Abstract

Autosomal dominant polycystic kidney disease (ADPKD) is the most common hereditary kidney disorder, but kidneys are not the only organs involved in this systemic disorder. Individuals with the condition may display additional manifestations beyond the renal system, involving the liver, pancreas, and brain in the context of cystic manifestations, while involving the vascular system, gastrointestinal tract, bones, and cardiac valves in the context of non-cystic manifestations. Despite kidney involvement remaining the main feature of the disease, thanks to longer survival, early diagnosis, and better management of kidney-related problems, a new wave of complications must be faced by clinicians who treated patients with ADPKD. Involvement of the liver represents the most prevalent extrarenal manifestation and has growing importance in the symptom burden and quality of life. Vascular abnormalities are a key factor for patients’ life expectancy and there is still debate whether to screen or not to screen all patients. Arterial hypertension is often the earliest onset symptom among ADPKD patients, leading to frequent cardiovascular complications. Although cardiac valvular abnormalities are a frequent complication, they rarely lead to relevant problems in the clinical history of polycystic patients. One of the newest relevant aspects concerns bone disorders that can exert a considerable influence on the clinical course of these patients. This review aims to provide the “state of the art” among the extrarenal manifestation of ADPKD.

## 1. Introduction

Autosomal dominant polycystic kidney disease (ADPKD) is the most prevalent hereditary kidney disorder, estimated to affect approximately 1 in every 1000 to 2500 individuals, based on data from multiple ethnicities (European, African, South Asian, Latino, and Finnish) [1]. The majority of ADPKD cases, around 78%, result from mutations in the *PKD1* gene, while 15% are attributed to mutations in the *PKD2* gene. However, a small portion of families exhibit mutations in genes such as *GANAB*, *ALG9*, or *DNAJB11* [2,3].

*PKD1* is situated on chromosome 16 (16p13.3) and codes for polycystin-1 (PC1), a large glycoprotein featuring multiple domains that undergo cleavage at a G protein-coupled receptor proteolytic site. Conversely, *PKD2*, located on chromosome 4 (4q21), encodes polycystin-2 (PC2), a member of the transient receptor potential family of calcium-regulated cation channels. Both PC1 and PC2 are present on primary cilia, antenna-like organelles crucial for mechano-transduction. It is hypothesized that PC1 and PC2 convey information from the external environment to the cell. Cystogenesis occurs when the concentration of PC1 or PC2 falls below a certain threshold, while higher levels of PC1 and PC2 inhibit cyst formation in a dose-dependent manner [4,5]. The ADPKD mutation database has cataloged over 2500 distinct mutations in *PKD1* and *PKD2*. The pace of disease advancement in individual patients is primarily determined by the specific causative gene mutation. Patients with *PKD2* mutations tend to develop fewer cysts and experience a slower decline in renal function as compared to individuals with *PKD1* mutations. Furthermore, truncating mutations in *PKD1* typically result in a more severe form of the disease than missense mutations [6].

Clinically, ADPKD is characterized by both cystic and non-cystic manifestations. The main clinical characteristic involves the formation and gradual enlargement of multiple fluid-filled cysts scattered throughout the renal parenchyma. This leads to a gradual decline in renal function over several decades, often resulting in end-stage renal disease (ESRD) around or after the sixth decade of life. Hypertension is a frequently observed early manifestation in ADPKD, occurring in 50% to 70% of cases before any substantial decline in the glomerular filtration rate (GFR). The average age of onset for hypertension is around 30 years and appears to be more prevalent in patients with enlarged kidneys or reduced GFR [7,8].

Additional complications associated with renal cyst growth and expansion include urinary tract infections, concentrating defects, hematuria, nephrolithiasis, and acute or chronic flank and abdominal pain. Proteinuria is not a typical feature of the disease, but its presence as a manifestation of chronic kidney disease (CKD) could worsen the prognosis [9]. Although less common, renal cell carcinoma (RCC) can also be a complication of ADPKD [10].

The kidney, however, is not the only organ damaged in ADPKD: affected individuals might exhibit extrarenal manifestations, involving the liver and pancreas in the context of cystic manifestations, and involving the gastrointestinal tract, vascular system, bones, and cardiac valves in the context of non-cystic manifestations.

Further, by causing a reduction in glomerular filtration, ADPKD is also characterized by the usual complications of CKD, such as anemia, secondary hyperparathyroidism, metabolic bone disease, inadequate nutrition, and increased cardiovascular risk.

## 2. Pathogenesis

ADPKD is classified as a ciliopathy. Cilia are conserved structures found in various organisms, and mutations affecting primary cilia, known as immotile cilia, can lead to ADPKD. Immotile cilia possess an axoneme, which is a cytoskeletal scaffold composed of nine microtubule doublets. Intra-flagellar transport (IFT) allows for the movement of components into and out of cilia [6,11]. The axoneme is tethered to the cell through a basal body originating from the centriole [12].

Primary cilia, typically non-motile monocilia, are situated on the surface of differentiated cells that are not actively dividing. They often exhibit a “9 + 0” microtubule configuration, featuring nine pairs of microtubules located in the outer part of the cell without a central apparatus or dynein arms. Primary cilia act as sensory organelles, detecting extracellular signals and functioning as surface mechano- or chemoreceptors. They sense changes in osmolality, light, temperature, and gravity, and they play critical roles in development and tissue differentiation. Various essential receptors, such as sonic hedgehog (SHH), epidermal growth factor receptor (EGFR), and platelet-derived growth factor receptor (PDGFR), are expressed on the surface of primary cilia. Signaling pathways associated with primary cilia include calcium, SHH, Wnt, mechanistic target of Rapamycin (mTOR), Janus kinase/signal transducer and activator of transcription (JAK/STAT), and mitogen activated protein kinase (MAPK), which play crucial roles in growth, distinctiveness, control of the cell cycle, programmed cell death, tissue homeostasis, and the alignment of cell orientation [13,14,15,16,17,18,19].

ADPKD belongs to a category of conditions referred to as primary ciliopathies, which encompass a range of syndromes and diseases involving multiple systems. Other examples of primary ciliopathies include Jeune syndrome, nephronophthisis, and Bardet–Biedl syndrome [11].

Primary cilia function as cellular antennae, responding to external stimuli and converting them into intracellular signals to regulate cellular functions. These stimuli can include physical stresses such as flow and pressure, as well as chemical substances like ligands, growth factors, and morphogens. The primary cilium possesses mechanosensing abilities, allowing it to detect fluid flow. PC1 and PC2 are among the ciliary proteins responsible for mechanosensing [20,21].

PC1 is a receptor-like protein featuring an extensive extracellular N terminus, 11 membrane-spanning domains, and a brief cytoplasmic C terminus. While PC1 exhibits high expression in fetal renal tissue, its expression is subdued in adult tissue. It is found in the cilium, plasma membrane, and adhesion complex in polarized epithelial cells, suggesting its involvement in protein–protein interactions, cell–cell adhesion, and cell–matrix interactions [22,23,24]. PC2 is an integral six-transmembrane protein with intracellular N and C termini. It functions as a nonselective calcium-permeable transient receptor potential channel [25,26]. While PC2 is co-localized with PC1 both in cilium and the plasma membrane, a significant part of PC2 is found inside the cell, where it plays a role in releasing calcium from intracellular stores [27,28,29]. PC1 and PC2 establish a complex by means of their C-terminal tails, contributing to intracellular calcium regulation. Activation of the PC1-PC2 channel complex occurs in response to ciliary bending, leading to signal transduction triggered by chemical or mechanical stimuli [21]. PC2, together with inositol 1, 4, 5-triphosphate receptor (IP3R) and ryanodine receptor, indirectly regulates cytoplasmic calcium levels [29,30,31,32,33,34,35].

The complete pathological mechanisms of ADPKD are not yet fully understood, but the loss of function of PC1 and/or PC2 proteins contributes to its pathogenesis through various signaling pathways [33]. The PC1-PC2 complex acts as a transient receptor potential channel involved in maintaining intracellular calcium homeostasis and calcium release. Interruption of the interaction of PC1/PC2 leads to decreased intracellular calcium levels, resulting in upregulated cyclic adenosine monophosphate (cAMP) signaling and increased cell proliferation [36,37,38,39]. PC1 and PC2 engage with numerous pathways, including the mTOR and JAK-STAT pathways, to inhibit cell growth. PC2 also reduces cell proliferation through its binding to eukaryotic translation elongation initiation factor 2a (eIF2a) and pancreatic ER-resident eIF2a kinase [40,41,42]. PC1 and PC2 also play roles in the Wnt signaling pathway, which regulates cell proliferation, specialization, and orientation. Defects in planar cell polarity can trigger renal tubule expansion and cyst formation [43].

*PKD1* gene mutations more often result in a more severe form of ADPKD and appear first compared to mutations in *PKD2* [44,45]. In the ADPKD mutation database, the position of each mutation determines the gravity. Truncating mutations often lead to more severe phenotypes than nontruncating mutations [46,47]. ADPKD exhibits genetic dominance at the organismal level but operates with a recessive mechanism at the cellular level. Cysts specifically develop in certain kidney tubules and hepatic bile ducts. Nevertheless, within adult tissues, both alleles of the mutated polycystic gene experience a recessive loss of function, leading to the development of cysts in a subset of tubular epithelial cells. This phenomenon is illustrated by the “second hit” hypothesis, where the presence of an inherited *PKD1* or *PKD2* mutation leads to cysts only if the remaining normal copy of the gene acquires a somatic mutation [48,49,50].

## 3. Extrarenal Cystic Manifestations

### 3.1. Liver

The liver is the most common organ involved in ADPKD, with liver cysts present in more than 90% of ADPKD patients (Figure 1). The current literature arbitrarily defines polycystic liver disease (PLD) as the existence of over 20 liver cysts. However, in patients with a family history of PLD, the finding of four or more cysts is sufficient to diagnose PLD [51]. PLD development is linked to structural alterations in the biliary tree, resulting in cyst formation, that normally occur early in the disease process [52,53,54]. Symptoms are due to cysts’ growth and typically occur in adulthood. [55]. The cysts’ distribution could be focal or diffuse to the whole organ. [56] The prevalence of PLD has increased due to factors such as a longer life expectancy and improved kidney survival. PLD associated with ADPKD is genetically distinct from isolated PLD but follows a similar clinical course, featuring hepatomegaly caused by multiple cysts while preserving liver function [56,57,58]. Several risk factors play a role in the formation of more severe PLD. These include the severity of the renal lesion, female sex, exogenous estrogen exposure, and repeated pregnancies. Although both males and females with ADPKD have a similar overall prevalence of PLD, cysts in females tend to be bigger and to appear earlier. The growth of hepatic cysts in females may be accelerated due to steroid hormones, as evidenced by the association between postmenopausal estrogen and the selective enlargement of hepatic cysts and parenchyma [59,60,61,62]. Furthermore, studies have shown that 58% of females older than 48 years with severe PLD experience a regression in liver volume, while the liver continues to enlarge in males. Therefore, women with severe polycystic liver disease should refrain from hormone replacement therapies and contraceptives containing estrogen [63,64,65]. The risk of developing severe PLD is not influenced by the ADPKD genotype but is associated with the gravity of renal disease [59]. Symptoms in PLD depend on the number, size, location, and distribution of cysts, which contribute to hepatomegaly. Liver cysts often remain asymptomatic and rarely lead to hepatic function impairment. However, approximately 20% of these patients experience symptomatic PLD. Symptoms are typically associated with progressive liver enlargement and may include pain, dyspepsia, gastroesophageal reflux, and, when the size compresses the portal vein, can lead to portal hypertension and ascites. Extensive enlargement of the liver can exert pressure on the nearby gastrointestinal tract, blood vessels, and diaphragm, resulting in diverse symptoms. Moreover, complications like cyst hemorrhage, rupture, or infection may give rise to additional symptoms. Highly symptomatic PLD has become more common due to the increased life expectancy of ADPKD patients [65,66,67,68,69,70,71].

The most used clinical classifications of PLD were written by Gigot and Schnelldorfer [72,73].

The Gigot classification serves to differentiate between phenotypes by considering the number, the size, and the amount of the liver involved [72]. Gigot classification relies on imaging studies but has the limits of not including symptoms and not evaluating the advancement of the condition.

The Schnelldorfer classification incorporates factors such as the quantity and size of cysts, the volume of remaining liver parenchyma, the input and output of previously preserved liver segments, and the presence of symptoms [73].

For the assessment of PLD symptoms, two specific questionnaires have been created and validated: POLCA (PLD complaint-specific assessment) [74] and PLD-Q (PLD questionnaire) [75]. Following completion, the total score is determined by summing individual symptom scores, with a higher total score indicating a greater disease burden. Patients actively participated in the development of PLD-Q, unlike the development of POLCA, resulting in distinct sets of items. Nevertheless, both questionnaires are applicable for gauging disease burden and evaluating changes in symptom burden post-treatment. Since treatment is recommended exclusively for symptomatic patients with hepatomegaly, both instruments can serve as novel clinical endpoints [76]. Liver volume, measured through CT or MRI volumetry using (semi-)automatic software, acts as a prognostic marker impacting both symptom burden and quality of life [76]. Two classifications are available to differentiate between mild, moderate, and severe phenotypes based on htTLV [57,77]. In the classification introduced by Hogan MC et al., mild PLD is arbitrarily defined as height-adjusted total liver volume (htLV) < 1000 mL/m, while htLV between 1000 and 1800 mL/m is considered moderate PLD. They designate severe PLD as htLV > 1800 mL/m, aligned with the plateauing of the height-adjusted liver parenchymal volume (htLPV) at this cutoff point. This point approximately corresponds to height-adjusted liver cyst volume (htLCV) > 700 mL/m [58]. In the classification put forth by Kim H et al., they categorize patients with PLD into three groups based on their htLV: no or mild PLD (htTLV < 1600 mL/m), moderate PLD (1600 ≤ htLV < 3200 mL/m), and severe PLD (htLV ≥ 3200 mL/m) [77].

In the management of PLD, therapeutic approaches are tailored to the individual patient based on their specific needs. One common complaint among patients is related to symptoms caused by progressive hepatic enlargement. The only medical therapy that has been shown to reduce liver dimension and improve quality of life in these symptomatic patients is the use of somatostatin analogues (SAs). Somatostatin hinders the production of cyclic adenosine monophosphate (cAMP) in cystic cholangiocytes. This compound is excessively produced in PLD and contributes to cellular growth and cystic fluid production. By decreasing fluid secretion and cell proliferation, SAs can reduce liver volume. Studies have demonstrated that six months of lanreotide injections can lead to a decrease in liver volume, and extending the treatment for an additional six months may lead to the stabilization of hepatic volume. Conversely, the use of vasopressin receptor 2 (V2) antagonists does not seem to reduce liver volume [78,79,80].

When symptoms are caused by a dominant cyst, patients may be eligible for aspiration sclerotherapy (AS). AS is a less invasive procedure that entails puncturing the cyst with radiological guidance, aspirating the cyst fluid, and then injecting a sclerosing agent. This process aims to reduce the cyst volume [81].

In cases where symptoms are caused by multiple larger cysts and if these cysts are accessible, fenestration can be considered. Fenestration involves both aspirating and surgically deroofing liver cysts. However, this approach is suitable only when the distribution of cysts permits access [76].

If the distribution of cysts does not allow for AS or fenestration, segmental hepatic resection may be considered. This procedure is reserved for cases of symptomatic and severe hepatomegaly, where a few liver segments are significantly affected by multiple cysts while other segments are less affected. It is important to note that segmental hepatic resection carries a higher risk of perioperative complications and may complicate future liver transplants due to the formation of adhesions [76].

Liver transplantation is the only curative treatment for severe and advanced cases of PLD. However, only a minority of patients will qualify for this intervention. Patients with massive hepatomegaly who suffer from severe malnutrition, low serum albumin levels, sarcopenia, or severe and recurrent complications such as cyst infections or portal hypertension may be considered for liver transplantation. The Model for End-Stage Liver Disease (MELD) score, which assesses the three-month prognosis in patients with liver failure, is an important tool for selecting patients for liver transplantation. In cases of ADPKD with severe renal impairment, combined liver–kidney transplantation should be considered [76].

It is worth noting that the choice of therapy should be made in consultation with healthcare professionals who specialize in the management of PLD, taking into consideration the individual patient’s condition, symptoms, and other factors.

### 3.2. Pancreas and Spleen

In ADPKD, the pancreas can be affected by the development of cysts in around 7% to 36% of affected patients. The loss of the proteins PC1 or PC2 is responsible for cysts’ formation in a mouse model. *PKD2* mutations are more prone (5.9 times) to develop those cysts. Cilia, which are cellular structures important for various functions, including tissue organization, are found exclusively in islet and ductal cells in pancreatic tissues. Their absence or disorganization can lead to abnormalities in the pancreatic ducts, loss of acinar cells, polarity defects, and dysregulated insulin secretion. As in other pancreatic pathologies, in the rare event of severe dysfunction of the endocrine or exocrine pancreas, therapy with insulin or replacement pancreatic enzymes, respectively, is indicated [82,83,84,85]. In a small percentage of cases, the pancreatic cysts found turned out to be IPMNs; these require periodic radiological monitoring and surgical removal in case of malignant evolution. [86].

In terms of splenic involvement in ADPKD, the incidence of splenic lesions appears to be similar between individuals with ADPKD and those without ADPKD (7% vs. 5%). The prevalence of splenic lesions does not appear to be influenced by the type of mutated gene. However, the median spleen volume (SV) is notably larger in patients with ADPKD compared to the general population (236 mL vs. 176 mL). Height-adjusted SV (htSV) is closely linked to height-adjusted total kidney volume (htTKV), indicating a relationship between kidney and spleen size in ADPKD patients. It did not seem to be associated with PLD or cyst complications [87].

Additional investigation is required to comprehensively grasp the mechanisms and clinical implications of pancreatic and splenic involvement in ADPKD.

### 3.3. Brain

Arachnoid cysts (ACs) are indeed another extrarenal manifestation that can occur in patients with ADPKD. Most ACs are considered congenital and asymptomatic. However, their prevalence is greater in ADPKD patients compared to non-ADPKD patients [88,89]. The reported prevalence of ACs in ADPKD ranges from 4.8% to 12.8%, whereas in the broader population, it is around 0.5% to 1.1%. The main sites of these cystic lesions in ADPKD are the middle cranial fossa (61%) and the posterior cranial fossa (39%), with no differences in ADPKD patients [90].

*PKD1* mutations seem to have a higher risk and earlier age of diagnosis of ACs compared to those with *PKD2* mutations. However, most ACs in ADPKD patients are neurologically asymptomatic and rarely associated with subdural hematoma.

The dimensions of ACs can differ, ranging from tiny to large cysts that encompass a significant portion of the cranial cavity. However, unlike kidney cysts, the alterations in volume for ACs are minor and do not correspond to the consistent and more pronounced expansion observed in kidneys. Furthermore, it appears that arachnoid cysts may be more prevalent in the advanced stages of ADPKD.

While most ACs are asymptomatic, they can rarely be associated with subdural hematoma. The presence of ACs may be influenced by the specific genetic mutation involved in ADPKD, with a higher risk observed in individuals with *PKD1* mutations [89,90]. Moreover, most ACs are asymptomatic and do not require treatment, but they can rarely be associated with subdural hematoma. When symptomatic, ACs can present with headache, cranial nerve dysfunction (visual system, facial palsy, and hearing loss), and nausea/vomiting. In these symptomatic cases, the surgical removal of ACs is indicated [91].

### 3.4. Other Involvements

In addition to the more commonly known manifestations of ADPKD, such as liver and kidney cysts, there are several other extrarenal manifestations that can occur in ADPKD patients.

Arachnoid membrane cysts, although rare, can occur in approximately 8% of ADPKD patients. These cysts are typically asymptomatic and found incidentally, but they may increase the risk for subdural hematomas, which are collections of blood between the brain and its outermost covering.

Spinal meningeal diverticula, which are outpouchings of the spinal meninges, can occur more frequently in ADPKD patients. However, they rarely present with intracranial hypotension, which is a condition characterized by low cerebrospinal fluid pressure due to a leak [92,93].

Cysts in the seminal vesicles are observed in approximately 40% of male ADPKD patients. While they are rarely directly responsible for infertility, they can contribute to defective sperm motility, leading to the same result. Prostate median cysts near the ejaculatory ducts have also been associated with ADPKD [94].

There is evidence suggesting a higher prevalence of bronchiectasis, a condition characterized by damage and the widening of the airways in the lungs, in ADPKD patients [95].

Colonic diverticula, which are small pouches that develop in the colon, as well as in the abdominal wall and abdominal hernias, have been documented to occur more frequently in patients with ADPKD [96].

Contrary to some reports, ovarian cysts are not associated with ADPKD [97].

It is worth noting that additional research is necessary to gain a comprehensive understanding of the prevalence, clinical consequences, and underlying mechanisms of these extrarenal manifestations in ADPKD.

## *4.* Extra-Renal Non-Cystic Manifestations

### 4.1. Vascular

Patients with ADPKD are known to have vascular abnormalities, including dissections and aneurysms in various large arteries throughout the body. Among the vascular complications, intracranial aneurysms (IAs) are the most prevalent in individuals with ADPKD (Figure 2). The existence of these vascular irregularities has given rise to the hypothesis that polycystins play a role in maintaining vascular integrity [98,99]. It is hypothesized that polycystins play a role in the development and maintenance of the vascular system. In a mouse model, impairment of *Pkd1* and *Pkd2* function influences endothelial cell responses to fluid shear stress, resulting in the diminished release of nitric oxide. This has been proposed as a potential contributor to hypertension in these patients. [21,100,101,102,103,104,105]. Interestingly, while the impairment of *Pkd1* and *Pkd2* yields similar phenotypes in endothelial cells, they appear to have antagonistic effects in vascular smooth muscle cells (VSMc). Deleting *Pkd1* in VSMc reduces arterial myogenic tone, and knocking down *Pkd2* in *Pkd1*-deficient arteries restores the myogenic response. The precise mechanisms through which specific PKD mutations make individuals prone to a vascular phenotype are still not fully understood [104,105,106].

The incidence of IAs in ADPKD is reported to be 4–5 times greater than in the general population, with estimates ranging from 9% to 12% compared to 2% to 3%. The IAs in these patients are usually more numerous and rupture 10 years earlier than in the general population [88,100]. There are modifiable and nonmodifiable risk factors associated with IAs’ rupture. Nonmodifiable factors include female gender, older age, personal or family history of aneurysm or subarachnoid hemorrhage (SAH), and certain ethnicities. Factors that can be modified include smoking habits, hypertension, and excessive alcohol consumption. Having a positive family history of IAs or SAH is a significant risk factor, and certain ethnic populations, such as Chinese, Japanese, or European, may be at elevated risk [107,108].

IAs, in ADPKD patients, are more frequently found on large-caliber arteries of the anterior circulation, mostly the internal carotid artery (ICA) and middle cerebral artery (MCA) [109].

It is important for ADPKD patients to undergo appropriate screening and monitoring for the presence of IAs to prevent potential complications such as rupture and SAH [110]. In patients with ADPKD, individuals with a positive family history have a higher prevalence of IAs (21.6%). Nevertheless, even among those with a negative family history, the prevalence remains elevated at 11%. Additionally, the prevalence of IAs tends to increase with age in ADPKD patients [111]. ADPKD patients also exhibit a higher frequency of anatomical variants in the arterial system, such as fenestrations, duplications, or azygos variants, compared to the general population. These variants may contribute to an increased susceptibility to developing aneurysms due to alterations in the arterial wall structure and turbulent blood flow. Fenestrations, in particular, can create turbulent flow and defects in the arterial segment, increasing the risk of aneurysm formation. Other arterial variants like duplications and azygos configurations may also promote aneurysm development through turbulent flow [88,112,113]. The rupture of intracranial aneurysms in ADPKD can lead to severe neurological complications and high morbidity and mortality rates, exceeding 50% of cases. Aneurysm size, location, and previous SAH are strong predictors of rupture [114]. The PHASES scoring system has been proposed to rate the risk of aneurysm rupture based on population, hypertension, age, aneurysm size and location, and previous SAH. This scoring system helps in prognostication and decision-making for appropriate management [115].

Time-of-flight magnetic resonance angiography (MRA) is the recommended screening approach for IAs as it does not require contrast agents. However, current guidelines do not universally recommend brain MRA for screening in ADPKD patients, suggesting its use primarily for those with a positive family history. Nevertheless, recent studies suggest that screening for IAs should be considered even in ADPKD patients without a known familial history to optimize patient management [88].

Management decisions for unruptured IAs in ADPKD are complex and involve several factors, including the patient’s age, overall health, aneurysm characteristics, and the feasibility of intervention. ADPKD patients have shown a greater incidence of iatrogenic complications such as hemorrhage, infarction, and dissection during aneurysm treatment compared to non-ADPKD patients. Conservative management is often appropriate for small asymptomatic aneurysms (<7 mm) in the anterior circulation, especially in patients with no history of SAH. Regular imaging follow-up is recommended initially, with longer intervals once aneurysm stability has been established. Lifestyle modifications, including smoking cessation, reducing alcohol consumption, and the strict control of blood pressure and dyslipidaemia, are important in minimizing the risk of aneurysm growth and rupture [107,116].

### 4.2. Heart

Arterial hypertension (AH) is indeed a common extrarenal manifestation of ADPKD, with an early onset typically occurring before the decline in renal function. AH in ADPKD is complex. Enlarged kidney cysts can exert pressure on kidney vessels, leading to regional ischemia and activation of the intrarenal renin-angiotensin-aldosterone system (RAAS) and the renal ortho-sympathetic nerve endings. Additionally, PC are proteins that constitute the structure of the vessel, so their alterations can lead to endothelial dysfunction and the impaired contractility of VSMc. However, persistent pressure burden and the loss of daily blood pressure decrease can contribute to the heightened occurrence of cardiovascular events in ADPKD [117,118]. The HALT PKD study demonstrated that strict control of blood pressure is linked to a slower progression of total kidney volume, no significant alteration in eGFR, a more pronounced decrease in left ventricular mass index, and a more substantial reduction in urinary albumin excretion. Monotherapy with an ACE inhibitor has been shown to be effective in achieving AH management in most ADPKD patients with CKD, making it a recommended first-choice therapy for young patients [118].

Valvular abnormalities are also observed in a notable portion (25–30%) of individuals with ADPKD. The most frequent irregularities are represented by mild mitral valve prolapse and aortic regurgitation, while mitral and/or tricuspid regurgitation are less frequent. These valve diseases in ADPKD may be attributed to generalized abnormalities in collagen or the extracellular matrix. The size of the aortic root and the pressure gradient across the aortic valve have been found to correlate with kidney volume adjusted for height, and the left ventricular septal wall thickness correlates with eGFR. The increased prevalence of mitral valve regurgitation may be influenced by hypertension, as sustained elevated blood pressure can increase the risk of mitral valve regurgitation. However, it is also suggested that mitral valve prolapse could be considered a characteristic manifestation of ADPKD, as it can occur in children and young adults independent of blood pressure [118,119,120,121,122,123]. Although mitral valve prolapse is a frequent complication in ADPKD, it rarely leads to severe cardiac dysfunction or significant clinical problems in polycystic patients. Therefore, routine screening echocardiography for mitral valve prolapse is not typically performed. In the rare cases in which the valvular disease becomes symptomatic, surgical treatment for valve correction is indicated. [124].

In addition to valvular abnormalities, other cardiovascular abnormalities have been associated with *PKD1* mutations in ADPKD patients, including left ventricular hypertrophy (LVH), arrhythmias, and dilated cardiomyopathy.

Comparing patients with advanced CKD, LVH is present in roughly 65% of ADPKD vs. 55% of non-ADPKD patients. It is worth noting that cardiovascular abnormalities in ADPKD are primarily limited to the left ventricle or atrium, with minimal changes observed in the right chambers. When possible, treatment with Angiotensin-Converting Enzyme Inhibitors (ACEis), angiotensin receptor blockers (ARBs), or mineralocorticoid receptor antagonists (MRAs) should be initiated. Mild tricuspid regurgitation has approximately the same prevalence in ADPKD as in non-ADPKD, with over 80% of individuals in both groups exhibiting this condition [121,122].

Notably, significant differences have been observed in the aortic root diameter among ADPKD patients based on the severity of cysts. Patients with a limited renal involvement (Mayo classes 1A/B) tend to have smaller aortic root diameters, while those with graver cysts (Mayo classes 1C-E) have larger aortic root diameters. This suggests a relationship between cyst severity and aortic root diameter in ADPKD [123].

Cardiovascular abnormalities related to ADPKD can vary among individuals and may not be present in all patients. Regular cardiovascular evaluations and monitoring are recommended to assess and manage these potential complications in ADPKD patients [124,125].

### 4.3. Bone

Bone disorders have emerged as a new area of study in the context of ADPKD, as they can have a significant impact on the clinical history of patients. The presence of primary cilia and polycystins in osteoblasts and osteocytes indicates their involvement in bone metabolism. PC1 acts as a mechanosensor and regulates osteoblastic gene transcription and bone cell differentiation. The dysregulation of *PKD1* expression within the skeletal framework may result in atypical bone growth and morphology, diminished bone mineral density, decreased cortical thickness, and the onset of osteopenia [126,127,128,129]. PC1 also contributes to chondrocyte and osteoblast differentiation and development and adipogenesis. Its inactivation due to a missense mutation can result in delayed bone formation. Animal models have shown that PC1 deficiency is associated with reduced mineralized bone content in both the calvaria and long bones, indicating a role in intramembranous and endochondral bone formation. This leads to decreased bone mineral density due to a decline in bone formation as opposed to an augmentation in bone resorption [126,127,130]. However, bone manifestations in humans with ADPKD are generally subtle or absent compared to animal models. It is important to consider that other syndromes with skeletal malformations can mimic ADPKD (examples: oral-facial-digital syndrome type 1 and serpentine fibula-polycystic kidney syndrome). In children with ADPKD, bone developmental defects have not been identified, and their growth and stature are typically normal [131].

Different types of *PKD* mutations result in the distinct responsiveness of cilia. Nontruncating *PKD2* mutations, associated with milder kidney disease, are linked to less responsive cilia compared to truncating *PKD2* or *PKD1* mutations. Truncating *PKD1* mutations result in osteoblasts exhibiting heightened cilia responsiveness and an expedited rate of mineralized matrix deposition [131].

Additionally, ADPKD patients with CKD stage 1–2 often have elevated levels of fibroblast growth factor 23 (FGF23), a hormone secreted by osteocytes. This leads to hypophosphatemia, renal phosphate wasting, and low circulating bone-specific alkaline phosphatase (BAP) levels. Interestingly, there is evidence of peripheral resistance to FGF23 in ADPKD patients, likely due to the disease and its klotho deficiency, considering that the tubular maximum phosphorus reabsorption per glomerular filtration rate surpasses the anticipated levels based on FGF23 exposure [127,131,132,133,134].

In summary, the observed bone defect in patients with ADPKD is discernibly different from the presentation observed in individuals with other etiologies of CKD. It is marked by an adynamic bone disorder, which refers to reduced bone turnover and mineralization. This bone defect manifests during the initial phases of CKD in individuals with ADPKD, even before the significant deterioration of renal function occurs. Nevertheless, even with the existence of the bone defect, individuals with ADPKD undergoing hemodialysis do not seem to exhibit a heightened risk of fractures when compared to non-ADPKD patients [127].

Since the study of bone abnormalities in ADPKD is relatively new, there are limited available data. Additional studies are required to investigate the changes occurring in the bones of ADPKD patients and to investigate the underlying aberrant regulatory pathways involved. Subsequent investigations will contribute to an enhanced comprehension of the specific bone lesions observed in ADPKD and their implications for patient care.

Here we provide a Table (Table 1) that summarizes medical and surgical treatments for extrarenal involvements in ADPKD patients.

## 5. Conclusions

ADPKD is a systemic disease that can involve the whole organism with cystic and non-cystic involvement.In patients with ADPKD, the kidney remains the main clinical feature, but liver, vascular, heart, and bone involvement could impact on patients’ quality of life.Liver volume is a prognostic marker and it impacts both symptom burden and quality of life.IA rupture is the most serious acute complication that can occur in ADPKD patients. Early detection and appropriate treatment are highly desirable.Although valvular abnormalities are common in ADPKD patients, they rarely lead to clinical problems; therefore, screening echocardiography is not compulsory.Bone defects in individuals with ADPKD align with adynamic bone disorder and it appears from the earliest stages of CKD.

## Figures and Tables

**Figure 1 ijms-25-02554-f001:**
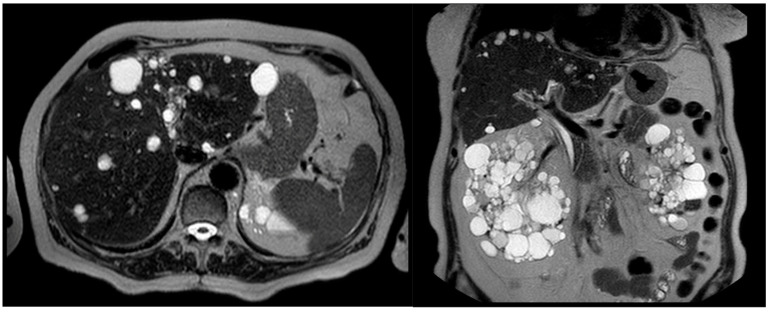
In the horizontal (**left**) and coronal (**right**) section of this computer tomography scan, the typical polycystic liver disease associated with ADPKD is shown.

**Figure 2 ijms-25-02554-f002:**
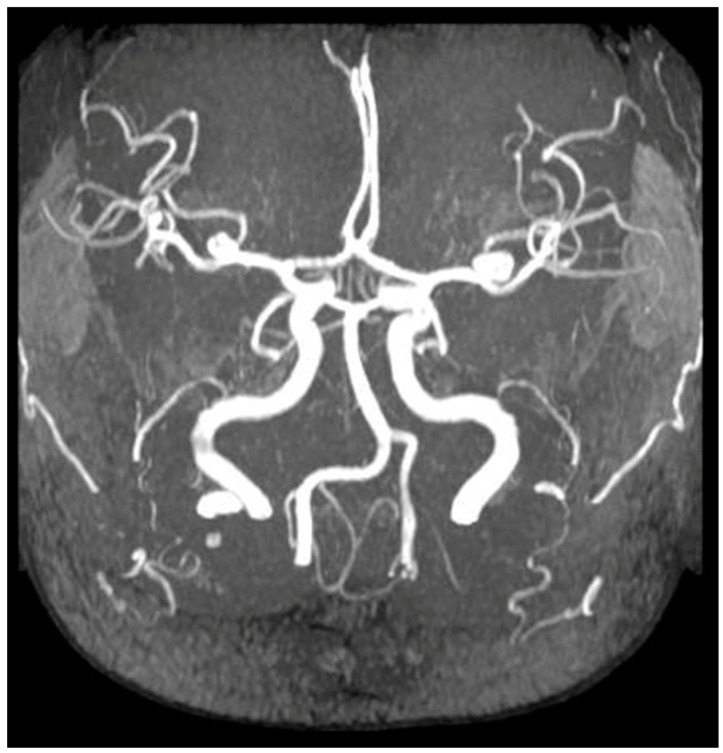
In this magnetic resonance phase contrast angiography, an intracranial aneurysm in a patient with ADPKD is shown.

**Table 1 ijms-25-02554-t001:** This table summarizes the main medical and surgical treatments for extrarenal involvement in ADPKD patients. IPMN: intraductal papillary mucinous neoplasm. ACs: arachnoid cysts. ACEi: angiotensin converting enzyme inhibitors. ARB: angiotensin receptor blockers. MRA: mineralocorticoid receptor antagonist.

Organ	Medical Therapy	Surgical Treatment
Liver	Somatostatin analogues	Aspiration sclerotherapy Fenestration Hepatic resection Liver transplant
Pancreas	Insulin Replacement pancreatic enzymes	Resection of malignant IPMN
Spleen	---	---
Brain	---	AC removal
Vascular	---	Evaluated on a case-by-case basis
Heart	ACEi, ARB, or MRA	Surgical correction
Bone	---	---

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
