# Peer review of "Autosomal Dominant Polycystic Kidney Disease: Extrarenal Involvement"

_ijms, 2024, doi:10.3390/ijms25052554_

Round 1

Reviewer 1 Report

Comments and Suggestions for Authors

Matteo and colleagues have studied over 140 pieces of literature related to Autosomal Dominant Polycystic Kidney Disease (ADPKD), focusing on the manifestations of this disease in organs other than the kidneys. Reviews in this field are relatively rare, and this paper provides an in-depth discussion of this area. The content is very rich, involving a wide range of organs, and the discussion is very scientifically rigorous, making it a paper worthy of publication. However, there are some minor issues:

1.     There are some inconsistencies in the paper, for example, the abstract states that the brain is considered a cystic manifestation, but in the main text, the brain is studied as a non-cystic manifestation. It is hoped that the authors will correct this;

2.     It should be stated which articles the data in Table 1 and Table 2 come from;

3.     The authors have provided 143 papers, but 80% of them are too outdated. It is recommended that the authors add papers from the past five years in this field to prove that these conclusions are still recognized by scientists;

4.     After discussing the manifestations of each organ, the authors briefly describe the current therapy. I suggest that the authors also summarize the current therapy into a table for the convenience of the readers;

5.     In my opinion, potential treatment methods are a reflection of the value of the article. In the paper by Cornec-Le Gall E that the authors cite, I believe the most valuable aspect is the author's assessment and prediction of potential treatment methods. I suggest that the authors could emulate this and provide an outlook on potential treatment methods for different organ manifestations.

Author Response

Dear editor and dear reviewers,

thank you for the opportunity to submit a revised version of our paper. Below we answer the reviewers’ criticisms point by point, the responses are indicated in blue. Our replies and changes to the text are tracked. We appreciate the time and effort of the reviewers for their thoughtful feedback. We feel that the revised version of this manuscript has benefited considerably from the review process. We hope that we have addressed each of the issues raised adequately. Please feel free to contact us with any additional questions or comments.

We look forward to your final decision on this paper.

Yours sincerely

Dr. Matteo Righini

MD, PhD

Response to reviewers comments concerning manuscript entitled: “Autosomal Dominant Polycystic Kidney Disease: extrarenal involvement” ID: ijms-2832349.

Reviewer 1:

There are some inconsistencies in the paper, for example, the abstract states that the brain is considered a cystic manifestation, but in the main text, the brain is studied as a non-cystic manifestation. It is hoped that the authors will correct this;

We have corrected according to your suggestion. We include “brain” as a cystic manifestation even in the main text

It should be stated which articles the data in Table 1 and Table 2 come from;

 We include the articles from which we extracted the tables.

The authors have provided 143 papers, but 80% of them are too outdated. It is recommended that the authors add papers from the past five years in this field to prove that these conclusions are still recognized by scientists;

Thanks to your tip, we revised our bibliography by selecting more recent articles and removing older ones, retaining only those essential for our discussion.

After discussing the manifestations of each organ, the authors briefly describe the current therapy. I suggest that the authors also summarize the current therapy into a table for the convenience of the readers;

According to your suggestion, we summarize the current therapy into a table. We provide a table that briefly describe treatment options for each organ issue.

In my opinion, potential treatment methods are a reflection of the value of the article. In the paper by Cornec-Le Gall E that the authors cite, I believe the most valuable aspect is the author’s assessment and prediction of potential treatment methods. I suggest that the authors could emulate this and provide an outlook on potential treatment methods for different organ manifestations.

Thanks to your comments we included treatment suggestions for each condition.

Reviewer 2 Report

Comments and Suggestions for Authors

See attached PDF

Comments on the Quality of English Language

See attached PDF

Author Response

Dear editor and dear reviewers,

thank you for the opportunity to submit a revised version of our paper. Below we answer the reviewers’ criticisms point by point, the responses are indicated in blue. Our replies and changes to the text are tracked. We appreciate the time and effort of the reviewers for their thoughtful feedback. We feel that the revised version of this manuscript has benefited considerably from the review process. We hope that we have addressed each of the issues raised adequately. Please feel free to contact us with any additional questions or comments.

We look forward to your final decision on this paper.

Yours sincerely

Dr. Matteo Righini

MD, PhD

Response to reviewers comments concerning manuscript entitled: “Autosomal Dominant Polycystic Kidney Disease: extrarenal involvement” ID: ijms-2832349.

Reviewer 2:

The reviewer said: “…the authors should provide the reader with the strategy

used to select appropriate references and how this review is different from the

content of chapters describing ADPKD in standard nephrology textbooks”

Thank you very much for the suggestion. We add a state at the bottom of our paper, describing the criterias we choose to select paper suitable for our state of art review.

The reviewer suggests: “…It would have been helpful if the submitted manuscript could

critically review the recent ADPKD literature relevant to extra-renal manifestations.

This would help to differentiate this submitted review from chapters on ADPKD in

textbooks.”

Thank you very much for your comment, this help us updating our review. We revised our bibliography considering a large number of more recent articles (from 2021 to 2023). We removed some older articles and replaced with more recents.

Point to correct and sentence to rephrase

We corrected the points that you suggested and rephrase the sentences according to your advice

Reviewer 3 Report

Comments and Suggestions for Authors

This is a review article that highlights the state of the art treatment protocols for autosomal dominant polycystic kidney disease. The aim is to raise awareness about the extrarenal complications associated with ADPKD, especially the organ involvement that can impact life. There are primary two genes that are responsible for disease development - PKD1 and PKD2. The severity of disease advancement depends on the gene mutation, PKD1 being more severe. It reports that ADPKD is the most prevalent hereditary kidney disorder, but I would like to know what demographic is the data based on? Is it country specific or global?

It affects the kidney by formation of multiple cysts in the renal parenchyma. It culminates into End Stage Renal Disease while hypertension is an early manifestation. There are other renal and extrarenal maifestations. Some extrarenal manifestations are due to the mechanism of PKD1 and PKD2 and some as a sequelae of renal malfunction. Please specify a reference for the extra renal manisfestations from 65-71?

The article explains the pathogenesis of ADPKD as a primary ciliopathy. The mechanism explains that multiple organ involvement.

There is a grammatical error in line 148 - 'occour' should be 'occur'

The paper then talks about polycystic liver disease and most importantly differentiates that besides the phenotypic difference the clinical presentation is not very different. I would like clarification about, in line 147 the paper reports that the presence of more than 20 cysts is defined as PLD, the Gigot and Schnelldorfer classification stage PLD with less number of cysts?

The paper is describing treatment modalities in liver and lists the medications/procedures available for PLD.

The severity of splenic lesions is not affected by the gene involved. But the size seems to be different in association with or without ADPKD. No treatment modalities have been mentioned. 

Other lesions, as reported, include cerebral and spinal cysts. The report highlights the importance of further investigation.

Apart from extrarenal cysts, other findings include vascular malformations and aneurysms. The paper also describes arterial hypertension and skeletal changes like modifications in trabecular pattern as well as bone turnover. 

The article describes pathologies in different organs associated with ciliopathies and corelates clinical presentations that are hypothesized and/or proven to be associated with ADPKD.

The article is a good summary of the pathogenic processes and helps understand the disease process that can encourage clinicians and researchers to investigate further. 

Comments on the Quality of English Language

Only one minor grammatical correction in line 148

Author Response

Dear editor and dear reviewers,

thank you for the opportunity to submit a revised version of our paper. Below we answer the reviewers’ criticisms point by point, the responses are indicated in blue. Our replies and changes to the text are tracked. We appreciate the time and effort of the reviewers for their thoughtful feedback. We feel that the revised version of this manuscript has benefited considerably from the review process. We hope that we have addressed each of the issues raised adequately. Please feel free to contact us with any additional questions or comments.

We look forward to your final decision on this paper.

Yours sincerely

Dr. Matteo Righini

MD, PhD

Response to reviewers comments concerning manuscript entitled: “Autosomal Dominant Polycystic Kidney Disease: extrarenal involvement” ID: ijms-2832349.

Reviewer 3:

It reports that ADPKD is the most prevalent hereditary kidney disorder, but I would like to know what demographic is the data based on? Is it country specific or global?

Thank you for your suggestion, we added a sentence concerning the demographic data

Please specify a reference for the extra renal manisfestations from 65-71?

We specified the reference in the considered paragraph

There is a grammatical error in line 148 - 'occour' should be 'occur'

We corrected according to your correction

I would like clarification about, in line 147 the paper reports that the presence of more than 20 cysts is defined as PLD, the Gigot and Schnelldorfer classification stage PLD with less number of cysts?

Thank you very much for your consideration, probably our paper wasn’t clear on the point. We rewrite the liver part according to your suggestion, stating that in patients with family history, 4 cysts are enough to describe a Polycystic Liver Disease.

Round 2

Reviewer 2 Report

Comments and Suggestions for Authors

I have noted a few minor corrections that can be made to the manuscript - see attached file

Comments on the Quality of English Language

Author Response

Dear editor and dear reviewers,
thank you for the opportunity to submit a revised version of our paper. Below we answer the reviewers’ criticisms point by point, the responses are indicated in blue. Our replies and changes to the text are tracked. We appreciate the time and effort of the reviewers for their thoughtful feedback. We feel that the revised version of this manuscript has benefited considerably from the review process. We hope that we have addressed each of the issues raised adequately. Please feel free to contact us with any additional questions or comments.

We look forward to your final decision on this paper. Yours sincerely

Dr. Matteo Righini MD, PhD

Response to reviewers comments concerning manuscript entitled: “Autosomal Dominant Polycystic Kidney Disease: extrarenal involvement” ID: ijms-2832349.

Reviewer 2:

2nd review – corrections to be made:

Correct the term polycystic to polycystic
Use IAs (not iA) as the abbreviation for intracranial aneurysm
Use IAs as the abbreviation (no iAs)
Line 522 – this is Table 3 and not Table 2. Correct the spelling of Table.
Rephrase first sentence in table to “This table summarises the main medical and surgical treatments for extrarenal involvement in ADPKD patients.”
Line 525 – correct spelling is mineralocorticoid (not mineralcorticoid)

We revised our paper according to your suggestions
